# Affinity-Aware Graph Networks

**Ameya Velingker**[*]
Google Research
ameyav@google.com

**Ali Kemal Sinop**[*]
Google Research
asinop@google.com

**Ira Ktena**
Google DeepMind
iraktena@google.com

**Petar Veličković**
Google DeepMind
petarv@google.com

**Sreenivas Gollapudi**
Google Research
sgollapu@google.com

## Abstract

Graph Neural Networks (GNNs) have emerged as a powerful technique for learning on relational data. Owing to the relatively limited number of message passing steps they perform—and hence a smaller receptive field—there has been significant interest in improving their expressivity by incorporating structural aspects of the underlying graph. In this paper, we explore the use of affinity measures as features in graph neural networks, in particular measures arising from random walks, including effective resistance, hitting and commute times. We propose message passing networks based on these features and evaluate their performance on a variety of node and graph property prediction tasks. Our architecture has low computational complexity, while our features are invariant to the permutations of the underlying graph. The measures we compute allow the network to exploit the connectivity properties of the graph, thereby allowing us to outperform relevant benchmarks for a wide variety of tasks, often with significantly fewer message passing steps. On one of the largest publicly available graph regression datasets, OGB-LSC-PCQM4Mv1, we obtain the best known single-model validation MAE at the time of writing.

## 1 Introduction

Graph Neural Networks (GNNs) constitute a powerful tool for learning meaningful representations in non-Euclidean domains. GNN models have achieved significant successes in a wide variety of node prediction [18, 31], link prediction [55, 52], and graph prediction [11, 50] tasks. These tasks naturally emerge in a wide range of applications, including autonomous driving [8], neuroimaging [36], combinatorial optimization [14, 34], and recommender systems [51], while they have enabled significant scientific advances in the fields of biomedicine [45], structural biology [23], molecular chemistry [42] and physics [3].

Despite the predictive power of GNNs, it is known that the expressive power of standard GNNs is limited by the 1-Weisfeiler-Lehman (1-WL) test [47]. Intuitively, GNNs possess at most the same power in terms of distinguishing between non-isomorphic (sub-)graphs, while having the added benefit of adapting to the given data distribution. For some architectures, two nodes with different local structures have the same computational graph, thus thwarting distinguishability in a standard GNN. Even though some attempts have been made to address this limitation with higher-order GNNs [32], most traditional GNN architectures fail to distinguish between such nodes.

A common approach to improving the expressive power of GNNs involves encoding richer structural/positional properties. For example, distance-based approaches form the basis for works such as

---

[*]Equal contribution

37th Conference on Neural Information Processing Systems (NeurIPS 2023).

Position-aware Graph Neural Networks [52], which capture positions/locations of nodes with respect to a set of anchor nodes, as well as Distance Encoding Networks [29], which use the first few powers of the normalized adjacency matrix as node features associated with a set of target nodes.

Here, we take an approach that is inspired by this line of work but departs from it in some crucial ways: we seek to capture both *distance* and *connectivity* information using general-purpose node and edge features without the need for specifying any anchor or target nodes.

**Contributions**: We propose the use of *affinity metrics* as features in a GNN. Specifically, we consider statistics that arise from random walks in graphs, such as *hitting time* and *commute time* between pairs of vertices (see Sections 3.1 and 3.2). We present a means of incorporating these statistics as scalar edge features in a message passing neural network (MPNN) [15] (see Section 3.4). In addition to these scalar features, we present richer vector-valued *resistive embeddings* (see Section 3.3), which can be incorporated as node or edge feature vectors in the network. Resistive embeddings are a natural way of embedding each node into Euclidean space such that the squared $L^2$-distance between nodes recovers the commute time. We show that such embeddings can be incorporated into MPNNs, *even for larger graphs*, by efficiently approximating them using sketching and dimensionality reduction techniques and prove a novel additive approximation for hitting time (see Section 4).

Moreover, we evaluate our networks on a number of benchmark datasets of diverse scales (see Section 5). First, we show that our networks outperform other baselines on the PNA dataset [9], which includes 6 node and graph algorithmic tasks, showing the ability of affinity measures to exploit structural properties of graphs. We also evaluate the performance on a number of graph and node tasks for datasets in the Open Graph Benchmark (OGB) collection [21], including molecular and citation graphs. In particular, our networks with scalar effective resistance edge features achieve the state of the art on the OGB-LSC PCQM4Mv1 dataset, which was featured in a KDD Cup 2021 competition for large scale graph representation learning.

Finally, we provide intuition for why affinity-based measures are fundamentally different from aforementioned distance-based approaches (see Section 3.5) and bolster it with detailed theoretical and empirical results (see Appendix D) showing favorable results for affinity-based measures.

## 2   Related Work

Our work builds upon a wealth of graph theoretic and graph representation learning works, while we focus on a supervised, inductive setting.

Even though GNN architectures were originally classified as spectral or spatial, we abstain from this division as recent research has demonstrated some equivalence of the graph convolution process regardless of the choice of convolution kernels [2, 7]. Spectrally-motivated methods are often theoretically founded on the eigendecomposition of the graph Laplacian matrix (or an approximation thereof) and, hence, corresponding convolutions capture different frequencies of the graph signal. Early works in this space include ChebNet [10] and its more efficient 1-hop version by Kipf et al. [24], which offers a linear function on the graph Laplacian spectrum. Levie et al. [28] proposed CayleyNets, an alternative rational filter.

Message passing neural networks (MPNNs) [15] perform a transformation of node and edge representations before and after an arbitrary aggregator (e.g. *sum*). Graph attention networks (GATs) [44] aimed to augment the computations of GNNs by allowing graph nodes to "attend" differently to different edges, inspired by the success of transformers in NLP tasks. One of the most relevant works was proposed by Beaini et al. [4], i.e. directional graph networks (DGN). DGN uses the gradients of the low-frequency eigenvectors of the graph Laplacian, which are known to capture key information about the global structure of the graph and prove that the aggregators they construct using these gradients lead to more discriminative models than standard GNNs according to the 1-WL test. Prior work [32] used higher-order ($k$-dimensional) GNNs, based on $k$-WL, and a hierarchical variant and proved theoretically and experimentally the improved expressivity in comparison to other models.

Other notable works include Graph Isomorphism Networks (GINs) [47], which represent a simple, maximally-powerful GNN over discrete-featured inputs. Hamilton et al. [18] proposed a method to construct node representations by sampling a fixed-size neighborhood of each node and then performing aggregation over it, which led to impressive performance on large-scale inductive benchmarks. Bouritsas et al. [6] use topologically-aware message passing to detect and count graph substructures,

while Bodnar et al. [5] propose a message-passing procedure on cell complexes motivated by a novel color refinement algorithm that proves to be powerful for molecular benchmarks. Meanwhile, Horn et al. [19] propose a layer based on persistent homology to incorporate global topological information.

**Expressivity.** Techniques that improve a GNN's expressive power largely fall under *three* broad directions. While we focus on the feature-based direction in this paper, we also acknowledge that it in no way compels the GNN to use the additional provided features. Hence, we briefly survey the other two, as an indication of the future research in affinity-based GNNs we hope this work will inspire.

Another avenue involves modulating the *message passing rule* to make advantage of the desired computations. Popular recent examples of this include DGN [4] and LSPE [12]. DGNs leverage the graph's Laplacian eigenvectors, but they do not merely use them as input features; instead, they define a directional *vector field* based on the eigenvectors, and use it explicitly to anisotropically aggregate neighbourhoods. LSPE features a "bespoke pipeline" for processing positional inputs.

The final direction is to modulate the *graph* over which messages are passed, usually by adding new nodes that correspond to desired substructures. An early proponent of this is the work of [32], which explicitly performs message passing over $k$-tuples of nodes at once. Recently, scalable efforts in this direction focus on carefully chosen substructures, e.g., junction trees [13], cellular complexes [5].

## 3 Affinity Measures and GNNs

### 3.1 Random Walks, Hitting and Commute Times

Let $G = (V, E)$ be a graph of vertices $V$ and edges $E \subseteq V \times V$ between them. We define several natural properties of a graph that arise from a random walk. A random walk on $G$ starting from a node $u$ is a Markov chain on the vertex set $V$ such that the initial vertex is $u$, and at each time step, one moves from the current vertex to a neighbor, chosen with probability proportional to the weight of outgoing edges. We will use $\pi$ to denote the stationary distribution of this Markov Chain. For random walks on weighted, undirected graphs, we know that $\pi_u = \frac{d_u}{2M}$, where $d_u$ is the weighted degree of node $u$, and $M$ is the sum of edge weights.

The *hitting time* $H_{uv}$ from $u$ to $v$ is defined as the expected number of steps for a random walk starting at $u$ to hit $v$. We can also define the *commute time* between $u$ and $v$ as $K_{uv} = H_{uv} + H_{vu}$, the expected round-trip time for a random walk starting at $u$ to reach $v$ and then return to $u$.

### 3.2 Effective Resistance

A closely related quantity is the measure of *effective resistances* in undirected graphs. This quantity corresponds to the effective resistance if the whole graph was replaced with a circuit where each edge becomes a resistor with resistance equal to the reciprocal of its weight. We will use $\mathsf{Res}(u, v)$ to denote the effective resistance between nodes $u$ and $v$. For undirected graphs, it is known that [30] the effective resistance is proportional to the commute time, $\mathsf{Res}(u, v) = \frac{1}{2M} K_{uv}$.

Our broad goal is to incorporate effective resistances and hitting times as edge features in an MPNN. In Section 3.4, we will show they can provably improve MPNNs' expressivity.

### 3.3 Resistive Embeddings

Effective resistances allow us to define the *resistive embedding*, a mapping that associates each node $v$ of a graph $G = (V, E, W)$, where $W$ are the non-negative edge weights, with an embedding vector. Before we specify the resistive embedding, we define a few terms. Let $L = D - A$ be the graph Laplacian of $G$, where $D \in \mathbb{R}^{n \times n}$ is the diagonal matrix containing the weighted degree of each node and $A \in \mathbb{R}^{n \times n}$ is the adjacency matrix, whose $(i, j)^{th}$ entry is equal to the edge weight between $i$ and $j$, if it exists; and 0 otherwise. Let $B$ be the $m \times n$ edge-node incidence matrix, where $|V| = n$ and $|E| = m$, defined as follows: The $i$-th row of $B$ corresponds to the $i$-th edge $e_i = (u_i, v_i)$ of $G$ and has a $+1$ in the $u_i$-th column and a $-1$ in the $v_i$-th column, while all other entries are zero. Finally we will use $C \in \mathbb{R}^{m \times m}$ to denote the conductance matrix, which is a diagonal matrix with $C_{ii}$ being the weight of $i^{th}$ edge. It is easy to verify that $B^T C B = L$. Even though $L$ is not invertible, its null-space consists of the indicator vectors for every connected component of $G$. For example, if $G$ is

connected, then $L$'s nullspace is spanned by the all-1's vector [2]. Hence, for any vector $x$ orthogonal to all-1's, $L \cdot L^\dagger x = x$, where $L^\dagger$ is the pseudo-inverse.

We can express effective resistance between any pair of nodes using Laplacian matrices [30] as $\mathsf{Res}(u, v) = (\mathbf{1}_u - \mathbf{1}_v)^T L^\dagger (\mathbf{1}_u - \mathbf{1}_v)$, where $\mathbf{1}_v$ is an $n$-dimensional vector specifying the indicator for node $v$. We are now ready to define the resistive embedding.

**Definition 3.1.** (Effective Resistance Embedding) $\mathbf{r}_v = C^{1/2} B L^\dagger \mathbf{1}_v$.

A key property is that the effective resistance between two nodes in the graph can be obtained easily from the distance between their corresponding embeddings (see proof in Appendix C):

**Lemma 3.2.** *For any pair of nodes $u, v$, we have $\|\mathbf{r}_u - \mathbf{r}_v\|_2^2 = \mathsf{Res}(u, v)$.*

One can easily check that any rotation of $\mathbf{r}$ also satisfies Lemma 3.2, since rotations preserve Euclidean distances; more generally, if $U$ is an orthonormal matrix, then $U\mathbf{r}$ is also a valid resistive embedding. This poses a challenge if we want to use the resistive embeddings as node or edge features: we want a way to enforce that a (G)NN using them will do so in a way that is invariant or equivariant to any rotations of the embeddings. In our current work, we rely on data augmentation: at every training iteration, we apply random rotations to the input ER embeddings.

**Remark.** While data augmentation is a popular approach for promoting invariant and equivariant predictions, it is only *hinting* to the network that such predictions are favourable. It is also possible, in the spirit of the geometric deep learning blueprint [7], to combine ER embeddings with an $O(n)$-equivariant GNN, which rigorously enforces rotational equivariance. A popular approach to building equivariant GNNs has been proposed by [39], though it focuses on the full Euclidean group $E(n)$ rather than $O(n)$. We leave this exploration to future work.

**Definition 3.3.** Let $\mathbf{p} := \sum_u \pi_u \mathbf{r}_u$ be the mean of effective resistance embedding.

We might view $\mathbf{p}$ as a "weighted mean"[3] of $\mathbf{r}$. We will define the hitting time radius, $H_{\max}$, of a given graph as the maximum hitting time between any two nodes:

**Definition 3.4** (Hitting Time Radius). $H_{\max} := \max_{u,v} H_{u,v}$.

We will need the following to bound the hitting times we computed:

**Lemma 3.5.** *For any node $u$, $\|\mathbf{r}_u - \mathbf{p}\|^2 \leq \frac{H_{\max}}{M}$.*

The proof follows easily from the fact that $\mathbf{p}$ is a convex combination of all $\mathbf{r}$'s and Jensen's inequality.

## 3.4 Incorporating Features into MPNNs

We reiterate that our main aim is to demonstrate (theoretically and empirically) that there are good reasons to incorporate affinity-based measures into GNN computations.

In the simplest instance, a method that improves a GNN's expressive power may compute additional *features* (positional or structural) which would assist the GNN in discriminating between examples it otherwise wouldn't (easily) be able to. These features are then appended to the GNN's inputs for further processing. For example, it has been shown that endowing nodes with a one-hot based *identity* is already sufficient for improving expressive power [33]; this was then relaxed to any randomly-sampled scalar feature by [38]. It is, of course, possible to create dedicated features that even count substructures of interest [6]. Further, the adjacency information can be factorised [37] or eigendecomposed [12] to provide useful structural embeddings for the GNN.

We will focus our attention on exactly this class of methods, as it is a lightweight and direct way of demonstrating improvements from these computations. Hence, our baselines will all be instances of the MPNN framework [15], which we will attempt to improve by endowing them with affinity-based features. We start by theoretically proving that these features indeed improve expressive power:

**Theorem 3.6.** *MPNNs that make use of any one of (a.) effective resistances, (b.) hitting times, (c.) resistive embeddings are strictly more powerful than the WL-1 test.*

---

[2]Throughout this section, we will assume that our graph is connected. However everything applies to disconnected graphs, too.

[3]Note that the average of all $\mathbf{r}_u$'s will be 0. If the graph is regular, then $\mathbf{p}$ will also be 0.

*Proof.* Since the networks in question arise from augmenting standard MPNNs with additional node/edge features, we have that these networks are at least as powerful as the 1-WL test.

In order to show that these networks are strictly *more powerful* than the 1-WL test, it suffices to show the existence of a graph for which our affinity measure based networks can distinguish between certain nodes that a standard GNN (limited by the 1-WL test) cannot.

We present an example of a 3-regular graph on 8 nodes in Figure 1. It is well-known that a standard GNN that is limited by the 1-WL test cannot distinguish any pair of nodes in a regular graph, as the computation tree rooted at any node in the graph looks identical. However, there are three isomorphism classes of nodes in the above graph (denoted by different colors), namely, $V_1 = \{1, 2\}$, $V_2 = \{3, 4, 7, 8\}$, and $V_3 = \{5, 6\}$.

We now show that GNNs with affinity based measures can distinguish between a node in $V_i$ and a node in $V_j$, for $i \neq j$. We note that the hitting time from $a$ to $b$ depends only on the isomorphism classes of $a$ and $b$. Thus, we write $r_{i,j}$ as the effective resistance between a node in $V_i$ and a node in $V_j$. Note that $r_{i,j} = r_{j,i}$, and it is easy to verify that:

$$r_{1,1} = 2/3, r_{2,2} = 15/28, r_{3,3} = 4/7$$
$$r_{1,2} = r_{2,1} = 185/336$$
$$r_{2,3} = r_{3,2} = 209/336.$$

Hence, it follows that in a message passing step of an MPNN that uses effective resistances, vertices in $V_1$, $V_2$, and $V_3$ will aggregate feature multisets $\{r_{1,1}, r_{1,2}, r_{1,2}\} = \{2/3, 185/336, 185/336\}$, $\{r_{2,1}, r_{2,2}, r_{2,3}\} = \{185/336, 15/28, 209/336\}$, and $\{r_{2,3}, r_{2,3}, r_{3,3}\} = \{209/336, 209/336, 4/7\}$, respectively, all of which are all distinct multisets. Hence, such an MPNN can distinguish nodes in $V_i$ and $V_j$, $i \neq j$ for a suitable aggregation function.

If, instead of effective resistance, we use hitting time features or resistive embeddings, our results still hold. This is because, as we showed previously, the effective resistance between nodes is a *function* of the two hitting times in either direction, as well as of the resistive embeddings of the nodes. In other words, if either hitting time features or resistive embeddings are used as input features for an MPNN, this MPNN would be able to compute the effective resistance features by applying an appropriate function (e.g., Lemma 3.2 for the case of resistive embeddings). Having computed these features, the MPNN can distinguish any two graphs that the MPNN with effective resistance features can. □

### 3.5 Effective Resistance vs. Shortest Path Distance

It is interesting to ask how effective resistances compare with shortest path distances (SPDs) in GNNs, given the plethora of recent works that make use of SPDs (e.g., [49, 52, 29]). The most direct comparison of our effective resistance-based MPNNs would be to use SPDs as edge features in the MPNNs. However, note that SPDs along graph edges are trivial (unlike effective resistances, which incorporate useful information about the global graph structure).

An alternative to edge features would be to use (a) SPDs to a small set of *anchor nodes* as features in an MPNN (e.g., P-GNN [52]) or (b) a dense featurization incorporating shortest paths between all pairs of nodes (e.g., the dense attention mechanism in Graphormer [49]). We remark that the latter approach typically incurs an $O(n^2)$ overhead, which our MPNN-based approach avoids.

We empirically compare our models to MPNNs that use approaches (a) and (b). Results on the PNA dataset show that our effective resistance-based GNNs outperform these approaches. Furthermore, we complement the empirical results with a theoretical result (see Appendix D) showing that under a limited number of message-passing steps, effective resistance features can allow one to distinguish structures that cannot be done using shortest path features. This provides some insight into why effective resistances can capture structure in GNNs

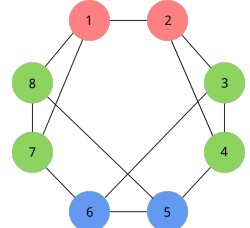

Figure 1: Degree 3 graph on 8 nodes, with isomorphism classes indicated by colors. While nodes of the same color are structurally identical, nodes of different colors are not. A standard GNN limited by the 1-WL cannot distinguish between nodes of different colors. However, affinity based networks that use effective resistances, hitting times, or resistive embeddings can distinguish every pair of such nodes.

that SPDs are unable to. We describe the result below. We point the reader to Appendix D for a proof of Theorem D.1.

# 4 Efficient Computation of Affinity Measures

In order to use our features, it is important that they be computable efficiently. In this section, we show how to compute or approximate the various random walk-based affinity measures. We share the results of our method on large-scale graphs in Section 5.4.

## 4.1 Reducing Dimensionality of Resistive Embeddings

Given their higher dimensionality, we might find it beneficial to use resistive embeddings as GNN features instead of the effective resistance. Now, the difficulty with using resistive embeddings directly as GNN features is the fact that the embeddings have dimension $m$, which can be quite large, e.g., up to $n^2$ for dense graphs. It was shown by [41] that one can reduce the dimensionality of the embed-

Table 1: $\log(MSE)$ on the PNA test dataset.

| Model | Avg score | Node tasks | | | Graph tasks | | |
|---|---|---|---|---|---|---|---|
| | | SSSP | Ecc | Lap feat | Conn | Diam | Spec rad |
| GAT | -1.730 | -2.213 | -1.935 | -2.644 | -0.618 | -1.430 | -1.538 |
| GCN | -1.592 | -2.283 | -1.978 | -1.698 | -0.618 | -1.432 | -1.541 |
| MPNN | -2.665 | -2.235 | -2.419 | -3.116 | -1.887 | -2.681 | -3.652 |
| MPNN (rand. feat.) | -2.490 | -2.136 | -1.808 | -3.873 | -1.696 | -2.614 | -2.813 |
| DGN (features) | -2.743 | -2.165 | -1.911 | -4.184 | -1.858 | -2.814 | -3.528 |
| DGN (features) + ER embed + HT | -2.938 | -2.360 | -2.949 | -3.689 | -1.744 | **-3.060** | -3.823 |
| ER GNN | -2.779 | -2.146 | -1.869 | -3.945 | **-1.962** | -2.940 | -3.811 |
| ER (node) embed. | -2.658 | -2.245 | -2.493 | -3.533 | -1.649 | -2.886 | -3.144 |
| ER (edge) embed. | -2.789 | -2.266 | -2.125 | -4.253 | -1.664 | -2.807 | -3.617 |
| Hitting Times | -2.816 | -2.189 | -1.904 | **-4.397** | -1.888 | -2.796 | -3.720 |
| **All ER features** | **-3.106** | **-2.789** | **-3.082** | -4.047 | -1.858 | -2.894 | **-3.962** |

ding while approximately preserving Euclidean distances. The idea is to use a random projection via a constructive version of the Johnson-Lindenstrauss Lemma:

**Lemma 4.1** (Constructive Johnson-Lindenstrauss). *Let $x_1, x_2, \ldots, x_n \in \mathbb{R}^d$ be a set of $n$ points; and let $\alpha_1, \alpha_2, \ldots, \alpha_m \in \mathbb{R}^n$ be fixed linear combinations. Suppose $\Pi$ is a $k \times d$ matrix whose entries are chosen i.i.d. from a Gaussian $N(0, 1)$ and consider $\widehat{x_i} := \frac{1}{\sqrt{k}} \Pi x_i$.*

*Then, it follows that for $k \geq C \log(mn)/\epsilon^2$, with probability $1 - o(1)$, we have $\| \sum_j \alpha_{i,j} \widehat{x}_j \|^2 = (1 \pm \epsilon) \| \sum_j \alpha_{i,j} x_j \|_2^2$ for every $1 \leq i \leq m$.*

Using triangle inequality, we can see that the inner products between fixed linear combinations are preserved up to some additive error:

**Corollary 4.2.** *For any fixed vectors $\alpha, \beta \in \mathbb{R}^n$, if we let $X := \sum_i \alpha_i x_i$, $\widehat{X} := \sum_i \alpha_i \widehat{x}_i$ and similarly $Y := \sum_i \beta_i x_i$, $\widehat{Y} := \sum_i \beta_i \widehat{x}_i$; then:*

$$\left| \langle X, Y \rangle - \langle \widehat{X}, \widehat{Y} \rangle \right| \leq \frac{\epsilon}{2} \left( \|X\|^2 + \|Y\|^2 \right).$$

Therefore, we can choose a desired $\epsilon > 0$ and $k = O(\log(n)/\epsilon^2)$ and instead use $\widehat{\mathbf{r}} : V \to \mathbb{R}^k$ as the embedding, where

$$\widehat{\mathbf{r}}_v = \frac{1}{\sqrt{k}} \Pi B L^\dagger e_v \tag{1}$$

for a randomly chosen $k \times d$ matrix $\Pi$ whose entries are i.i.d. Gaussians from $\mathcal{N}(0, 1)$. Then, by lemma 3.2 and lemma 4.1, we have that for every edge $(u, v) \in E$,

$$\|\widehat{\mathbf{r}}_u - \widehat{\mathbf{r}}_v\|_2^2 = (1 \pm 3\epsilon) \mathsf{Res}(u, v)$$

with probability at least $1 - \frac{1}{n^2}$.

So the computation of random embeddings, $\widehat{\mathbf{r}}$, requires solving $O((n + m) \log n/\epsilon^2)$ many Laplacian linear systems. By using one of the nearly linear time Laplacian solvers [26], we can compute the random embeddings in the near-linear time. Hence the total running time becomes $O\left((n + m) \log^{3/2} n \mathrm{poly} \log \log n/\epsilon^2\right)$.

## 4.2 Fast Computation of Hitting Times

Note that it is not clear whether there is a fast method for computing hitting times similar to commute times / effective resistances. The naive approach involves solving a linear system for each edge, resulting in a running time of at least $\Omega(nm)$, which is prohibitive. One of our technical contributions in this paper is a novel method for fast computation of hitting times. In particular, we will show how to use the approximate effective resistance embeddings, $\widehat{\mathbf{r}}$, to obtain an estimate for hitting times with additive error. Proofs can be found in Appendix C.

Let $\widehat{\mathbf{p}} := \sum_u \pi_u \widehat{\mathbf{r}}_u$. Just like $\widehat{\mathbf{r}}$ being an approximation of $\mathbf{r}$, $\widehat{\mathbf{p}}$ is an approximation of $\mathbf{p}$. Consider the following quantity:

$$\widehat{H}_{u,v} = 2M\langle \widehat{\mathbf{r}}_v - \widehat{\mathbf{r}}_u, \widehat{\mathbf{r}}_v - \widehat{\mathbf{p}}\rangle. \tag{2}$$

We will use this quantity as an approximation of $H_{u,v}$. In the following part, we will bound the difference between $H_{u,v}$ and $\widehat{H}_{u,v}$. Our starting point will be expressing $H_{u,v}$ in terms of the effective resistance embeddings.

**Lemma 4.3.** $H_{u,v} = 2M\langle \mathbf{r}_v - \mathbf{r}_u, \mathbf{r}_v - \mathbf{p}\rangle$ *where* $\mathbf{p} := \sum_u \pi_u \mathbf{r}_u$.

Given Lemma 4.3, we can establish the desired additive approximation property of $\widehat{H}_{u,v}$.

**Lemma 4.4.** $|\widehat{H}_{u,v} - H_{u,v}| \leq 3\epsilon H_{\max}$.

# 5 Experiments

As previously discussed, our empirical evaluation seeks to show benefits from endowing standard expressive GNNs with additional affinity-based features. All architectures we experiment with will therefore conform to the MPNN blueprint [15], which we describe in Appendix B. When relevant, we may also note results that a particular strong baseline (e.g., DGN [4], Graphormer [48]) achieves on a dataset of interest. Note that these baselines modulate the message passing procedure rather than appending features and are hence a different category to our method—their performance is provided for indicative reasons only. Where appropriate, we use "*DGN (features)*" to refer to an MPNN that uses eigenvector flows as additional edge features, without modulating the mechanism. We also use "ER (GNN)" to refer to the use of scalar effective resistance features.

## 5.1 PNA dataset

The first dataset we explore is the PNA dataset [9], which captures a multimodal setting. This consists of a collection of *node tasks*, i.e. (**1**) Single-source shortest paths, (**2**) Eccentricity and (**3**) Laplacian features, as well as *graph tasks*, i.e. (**4**) Connectivity, (**5**) Diameter and (**6**) Spectral radius. PNA dataset is a set of structured graph tasks, and it complements our other datasets. As we can see in Table 1, even adding a single feature, effective resistance (ER GNN), yields the best average score compared to other models. Using hitting times as edge features improve upon effective resistances. However once we combine all ER features, which include effective resistances, hitting times as well as node and edge embeddings, we get the best scores. On these structured tasks, we can see that the affinity based measures provide a significant advantage.

We also note that incorporating affinity measures as additional features to the "DGN (features)" MPNN model also provides improvement over the standard "DGN (features)" baseline. We have accordingly included the results of a model that uses (both node and edge) ER embeddings and hitting time features along with DGN features.

## 5.2 Small molecule classification: ogbg-molhiv

The ogbg-molhiv dataset is a molecular property prediction dataset comprised of molecular graphs without spatial information (such as atom coordinates). Each graph corresponds to a molecule, with nodes representing atoms and edges representing chemical bonds. Each node has an associated 9-dimensional feature, containing atomic number and chirality, as well as other additional atom features such as formal charge and whether the atom is in the ring or not. The goal is to predict whether a molecule inhibits HIV virus replication or not. Our results are given in Table 2.

On this dataset, effective resistances provide an improvement over the standard MPNN. We achieve the best performance using ER node embeddings and hitting times with random rotations. With these features, our network achieves $\mathbf{79.13}\% \pm 0.358$ test accuracy, which is close to DGN.

Note that we provide GSN [6], HIMP [13], and GR-WNN [35] as baselines. HIMP and GSN exploit structural/toplogical properties; the former is specifically designed for learning on molecular graphs, while the latter incorporates graph substructures. In addition to the comparison to GSN as a baseline, we additionally train an MPNN with substructure counts as additional features, namely counts of $k$-cycles ($k = 3, 4$) and $k$-paths ($k = 3$) as motifs (in the same vein as the approach of [6], obtaining a test ROC-AUC of $76.41\% \pm 0.42$ (note that this improves on the base MPNN test ROC-AUC of $74.67\% \pm 0.19$). Furthermore, if we also add affinity measures (in addition to the aforementioned substructure counts), we obtain a test ROC-AUC of $78.50\% \pm 0.51$. This suggests that affinity measures provide a performance or expressivity lift beyond that of substructure counts.

Table 2: Test % AUC-ROC on the Mol-HIV dataset. Our results are averaged over five seeds.

|  | MolHIV Test % ROC-AUC |
| --- | --- |
| GCN | $76.06 \pm 0.97$ |
| GIN | $75.58 \pm 1.40$ |
| MPNN | $74.67 \pm 0.19$ |
| DGN | $\mathbf{79.70} \pm 0.97$ |
| GSN (GIN + VN) | $77.99 \pm 1.00$ |
| HIMP | $78.80 \pm 0.82$ |
| GRWNN | $78.38 \pm 0.99$ |
| **ER GNN** | $77.75 \pm 0.426$ |
| **ER (node) embed. + HT (with random rotations)** | $\mathbf{\mathit{79.13}} \pm 0.358$ |

## 5.3 Multi-task molecular classification: ogbg-molpcba

The ogbg-molpcba dataset comprises molecular graphs without spatial information (such as atom coordinates). The aim is to classify them across 128 different biological activities. We follow the baseline MPNN architecture from [17], including the use of Noisy Nodes.

Mirroring the evaluation protocol of [17], Table 3 compares the performance of incorporating ER and hitting time (HT) features into the baseline MPNN models with Noisy Nodes, at various depths. What can be noticed is that models utilising affinity-based features are capable of reaching as well as exceeding peak test performance (in terms of mean average precision). However, what's important is the effect of these features at lower depths: it is possible to achieve comparable or better levels of performance with *half the layers*, when utilising ER or HT features. This result illustrates the potential benefit affinity-based computations can have on molecular benchmarks, especially when no spatial geometry is provided as input.

Table 3: ogbg-molpcba performance for various model depths. Best performance across all models is underlined. NN refers to Noisy Nodes.

|  | Test Mean Average Precision | | |
| --- | --- | --- | --- |
| Model | *4 layers* | *8 layers* | *16 layers* |
| MPNN [17] | $27.75\% \pm 0.20$ | $27.91\% \pm 0.22$ | $27.64\% \pm 0.25$ |
| MPNN + NN | $27.92\% \pm 0.11$ | $28.07\% \pm 0.14$ | $\mathbf{28.29}\% \pm 0.13$ |
| MPNN + NN + ER (ours) | $\mathbf{28.11}\% \pm 0.19$ | $28.27\% \pm 0.17$ | $28.28\% \pm 0.14$ |
| MPNN + NN + HT (ours) | $28.03\% \pm 0.15$ | $\underline{\mathbf{28.32}}\% \pm 0.13$ | $28.20\% \pm 0.19$ |

Table 4: ogbg-molpcba performance compared to other baselines.

| Model | Test Mean AP |
| --- | --- |
| GCN (w/ virtual node) | $24.24 \pm 0.34$ |
| GIN (w/ virtual node) | $27.03 \pm 0.23$ |
| DeeperGCN | $27.81 \pm 0.38$ |
| HIMP | $27.39 \pm 0.17$ |
| MPNN + NN + HT (ours) | $\underline{\mathbf{28.32}}\% \pm 0.13$ |

## 5.4 Scaling to larger graphs

Next, we present results on ogbn-arxiv and ogbn-mag, transductive datasets with large graphs.

### 5.4.1 Large citation network: ogbn-arxiv

Most expressive GNNs that rely on computation of structural features have not been scaled beyond small molecular datasets (such as the ones discussed in prior sections). This is due to the fact that computing them requires (time or storage) complexity which is at least quadratic in the graph size—making them inapplicable even for modest-sized graphs. This is, however, not the case for our proposed affinity-based metrics. We demonstrate this by scalably computing them on a larger-scale node classification benchmark, ogbn-arxiv (a citation network with the goal of predicting the arXiv category of each paper). ogbn-arxiv has 169,343 nodes and 1,166,243 edges, making quadratic approaches infeasible. Using the combinatorial multigrid preconditioner [27, 25], we constructed the effective resistances on this graph in an hour on a standard MacBook Pro 2019 laptop.

As MPNN models overfit this transductive dataset quite easily, the dominant approach to tackling it are graph attention networks (GATs) [44]. Accordingly, we trained a simple four-layer GAT on this dataset, achieving $72.02\% \pm 0.05$ test accuracy. This compares with $71.97\% \pm 0.24$ reported for a related attentional baseline on the leaderboard [54], indicating that our baseline performance is relevant.

ER embeddings on ogbn-arxiv need to be exceptionally high-dimensional to achieve accurate ER estimates ($\sim$11,000 dimensions), hence we were unable to use them here. However, incorporating *ER scalar* features into our GAT model yielded a statistically-significant improvement of $72.14\% \pm 0.03$ test accuracy. *Hitting time* features improve this result further to **72.25%** $\pm 0.04$ test accuracy. This demonstrates that our affinity-based metrics can yield useful improvements even on larger scale graphs, which are traditionally out of reach for methods like DGN [4] due to computational complexity limitations.

Reliable global leaderboarding with respect to ogbn-arxiv is difficult, as state-of-the art approaches rely either on privileged information (such as raw text of the paper abstracts), incorporating node labels as features [46], post-processing the predictions [22], or various related tricks [46]. With that in mind, we report for convenience that the current state-of-the-art performance for ogbn-arxiv without using raw text is $76.11\% \pm 0.09$ test accuracy, achieved by GIANT-XRT+DRGAT.

### 5.4.2 Heterogeneous citation network: ogbn-mag

We additionally present experiments on an even larger-scale network, ogbn-mag, a heterogeneous network derived from the Microsoft Academic Graph whose nodes consist of four types of entities (papers, authors, institutions, fields of study) and whose edges capture directed relations (e.g., a paper cites another paper, an author is affiliated with an institution, etc.). The task is to predict the venues of papers. The ogbn-mag dataset consists of 111,059,956 nodes and 1,615,685,872 edges.

As in the case of ogbn-arxiv, ER embeddings require an exceptionally high number of dimensions, given the large size of the network. Hence, we incorporated ER scalar features. As a baseline, we used a GraphSAINT [53] model that uses R-GCN aggregation [40], which reported a test accuracy of $47.51\% \pm 0.22$ in a relevant leaderboard entry. After incorporating ER scalar features into the same model, we obtained a statistically-signficant improvement of **47.99%** $\pm 0.23$ in test accuracy.

### 5.5 Large scale graph regression: OGB-LSC PCQM4Mv1

We finally include experimental results for one of the largest-scale publicly available graph regression tasks: the PCQM4Mv1 dataset from the OGB Large Scale Challenge [20]. PCQM4M is a quantum chemistry dataset spanning 4 million small molecules, with a task to predict the HOMO-LUMO gap, an important quantum-chemical property. It is anticipated that structural features such as ER could be of great help on this task, as the v1 version of it is provided without any structural information, and the molecule's geometry is assumed critical for predicting the gap. We report the single-model validation performance on this dataset, in line with previous works [17, 48, 1].

PCQM4Mv1 comprises molecular graphs which consist of bonds and atom types, and no 3D or 2D coordinates. We reuse the experimental setup and architecture from [17], with only one difference: appending the effective resistance to the edge features. Additionally, we compare against an equivalent model

Table 5: Single-model OGB-LSC PCQM4Mv1 results.

| Model | #Layers | Noisy Nodes | Validation MAE |
|---|---|---|---|
| MPNN [17] | 16 | Yes | $0.1249 \pm 0.0003$ |
| MPNN [17] | 50 | No | $0.1236 \pm 0.0001$ |
| Graphormer [48] | - | - | $0.1234$ |
| MPNN [17] | 50 | Yes | $0.1218 \pm 0.0001$ |
| MPNN + Conformers [1] | 32 | Yes | $0.1212 \pm 0.0001$ |
| MPNN + ER (ours) | 32 | Yes | $\mathbf{0.1197 \pm 0.0002}$ |

which uses molecular conformations estimated by RDKit as an additional feature. This gives us a baseline which leverages an explicit estimate of the molecular geometry.

Our results are summarised in Table 5. We once again see a powerful synergy of effective resistance-endowed GNNs and Noisy Nodes [17], allowing us to significantly reduce the number of layers (to 32) and outperform the 50-layer MPNN result in [17]. Further, we improve on the single-model performance of both the Graphormer [48] (which won the original PCQM4M contest after ensembling), and an equivalent model to ours which uses molecular conformers from RDKit. This illustrates how ER features can be competitive in geometry-relevant tasks even against features that inherently encode an estimate of the molecule's spatial geometry.

Lastly, we remark that, to the best of our knowledge, our result is the *best published single-model result* on the large-scale PCQM4M-v1 benchmark to date, and the only single model result with validation MAE under 0.120. We hope this will inspire future investigation on affinity-related GNNs for molecular tasks, especially in settings where spatial geometry is not reliably available.

## 6 Conclusions

In this paper, we proposed a message passing network based on random walk based affinity measures.

We believe that the comprehensive theoretical and practical results presented in our paper have solidified affinity-based computations as a strong component of a graph representation learner's toolbox. Our proposal carefully balances theoretical expressive power, empirical performance, and scalability to large graphs. While adding affinity measures as node/edge features provides the most direct route to incorporating them in a graph network, an interesting future direction would be to explore variants of GNN message functions that explicitly make use of *affinity-based computations*.

## Acknowledgments and Disclosure of Funding

We would like to specially thank Jonathan Godwin for his extensive support in setting up the codebase for large-scale GNNs, and Gabriele Corso for his support in setting up the PNA dataset. Additionally, we would like to thank Beatrice Bevilacqua, Wilfried Bounsi, Larisa Markeeva and Pete Battaglia for reviewing the paper prior to submission.

This research was funded by Google Research and Google DeepMind.

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

## A  Hyperparameters for PNA dataset.

In this section we provide the hyperparameters used for the different models on the PNA multitask benchmark. We train all models for 2000 steps and with 3 layers. The remaining hyperparameters for hidden size of each layer, learning rate, number of message passing steps (only valid for MPNN models), number of rotation matrices and same example frequency (when relevant) are provided in Table 6.

Table 6: Training hyperparameters for PNA dataset

| Model | #Hidden size | L. rate | #MP steps | #Rotation matrices | #Same Examples |
|---|---|---|---|---|---|
| GAT | 64 | $10^{-4}$ | - | - | - |
| GCN | 64 | $10^{-4}$ | - | - | - |
| DGN | 256 | $10^{-3}$ | - | - | - |
| MPNN | 256 | $10^{-3}$ | 2 | - | - |
| ER GNN | 128 | $10^{-3}$ | 2 | - | - |
| ER (node) embed. | 64 | $10^{-3}$ | 1 | - | - |
| ER (edge) embed. | 256 | $10^{-3}$ | 2 | - | - |
| ER (edge) embed. | 256 | $10^{-3}$ | 2 | - | - |
| All ER features | 256 | $10^{-4}$ | 2 | 23 | 9 |
| HT + ER (rand rot) | 512 | $10^{-4}$ | 2 | 23 | 4 |

## B  Details of MPNN Framework

As discussed previously, the architectures used in the experiments conform to the MPNN framework, which allows affinity measures to be added as additional node and edge features. We describe the details here for completeness.

Assume that our input graph, $\mathcal{G} = (\mathcal{V}, \mathcal{E})$, has node features $\mathbf{x}_u \in \mathbb{R}^n$, edge features $\mathbf{x}_{uv} \in \mathbb{R}^m$ and graph-level features $\mathbf{x}_\mathcal{G} \in \mathbb{R}^l$, for nodes $u, v \in \mathcal{V}$ and edges $(u, v) \in \mathcal{E}$. We provide encoders $f_n : \mathbb{R}^n \to \mathbb{R}^k$, $f_e : \mathbb{R}^m \to \mathbb{R}^k$ and $f_g : \mathbb{R}^l \to \mathbb{R}^k$ that transform these inputs into a latent space:

$$\mathbf{h}_u^{(0)} = f_n(\mathbf{x}_u) \qquad \mathbf{h}_{uv}^{(0)} = f_e(\mathbf{x}_{uv}) \qquad \mathbf{h}_\mathcal{G}^{(0)} = f_g(\mathbf{x}_\mathcal{G}) \tag{3}$$

Our *MPNN* then performs several message passing steps:

$$\mathbf{H}^{(t+1)} = P_{t+1}(\mathbf{H}^{(t)}) \tag{4}$$

where $\mathbf{H}^{(t)} = \left( \left\{ \mathbf{h}_u^{(t)} \right\}_{u \in \mathcal{V}}, \left\{ \mathbf{h}_{uv}^{(t)} \right\}_{(u,v) \in \mathcal{E}}, \mathbf{h}_\mathcal{G}^{(t)} \right)$ contains all of the latents at a particular processing step $t \geq 0$.

This process is iterated for $T$ steps, recovering final latents $\mathbf{H}^{(T)}$. These can then be *decoded* into node-, edge-, and graph-level predictions (as required), using analogous decoder functions $g_n$, $g_e$ and $g_g$:

$$\mathbf{y}_u = g_n(\mathbf{h}_u^{(T)}) \qquad \mathbf{y}_{uv} = g_e(\mathbf{h}_{uv}^{(T)}) \qquad \mathbf{y}_\mathcal{G} = g_g(\mathbf{h}_\mathcal{G}^{(T)}) \tag{5}$$

Generally, $f$ and $g$ are simple MLPs, whereas we use the MPNN update rule for $P$. It computes message vectors, $\mathbf{m}_{uv}^{(t)}$, to be sent across the edge $(u, v)$, and then aggregates them in the receiver nodes as follows:

$$\mathbf{m}_{uv}^{(t+1)} = \psi_{t+1}\left( \mathbf{h}_u^{(t)}, \mathbf{h}_v^{(t)}, \mathbf{h}_{uv}^{(0)} \right), \quad \mathbf{h}_u^{(t+1)} = \phi_{t+1}\left( \mathbf{h}_u^{(t)}, \sum_{u \in \mathcal{N}_v} \mathbf{m}_{vu}^{(t+1)} \right) \tag{6}$$

The message function $\psi_{t+1}$ and the update function $\phi_{t+1}$ are both MLPs. All of our models have been implemented using the jraph library [16].

We incorporate edge-based affinity features (e.g., effective resistances and hitting times) in $f_e$ and node-based affinity features (e.g., resistive embeddings) in $f_n$. Note that node-based affinity

features may also naturally be incorporated as edge features by concatenating the node features at the endpoints.

Occasionally, the dataset in question will be easy to overfit with the most general form of message function (see (6)). In these cases, we resort to assuming that $\psi$ factorises into an *attention mechanism*:

$$\mathbf{m}_{uv}^{(t+1)} = a_{t+1}\left(\mathbf{h}_u^{(t)}, \mathbf{h}_v^{(t)}, \mathbf{h}_{uv}^{(0)}\right)\psi_{t+1}\left(\mathbf{h}_u^{(t)}\right) \tag{7}$$

where the attention function $a$ is scalar-valued. We will refer to this particular MPNN baseline as a graph attention network (*GAT*) [44].

## C   Omitted Proofs

**Lemma 3.2.** *For any pair of nodes $u, v$, we have $\|\mathbf{r}_u - \mathbf{r}_v\|_2^2 = \mathsf{Res}(u, v)$.*

*Proof.*

$$\begin{aligned}
\|\mathbf{r}_u - \mathbf{r}_v\|_2^2 &= \|C^{1/2}BL_G^{-1}(\mathbf{1}_u - \mathbf{1}_v)\|_2^2 \\
&= (\mathbf{1}_u - \mathbf{1}_v)^T L^\dagger (B^T C B) L^\dagger (\mathbf{1}_u - \mathbf{1}_v) \\
&= (\mathbf{1}_u - \mathbf{1}_v)^T L^\dagger L L^\dagger (\mathbf{1}_u - \mathbf{1}_v) \\
&= (\mathbf{1}_u - \mathbf{1}_v)^T L^\dagger (\mathbf{1}_u - \mathbf{1}_v) = \mathsf{Res}(u, v). \qquad \square
\end{aligned}$$

**Corollary 4.2.** *For any fixed vectors $\alpha, \beta \in \mathbb{R}^n$, if we let $X := \sum_i \alpha_i x_i$, $\widehat{X} := \sum_i \alpha_i \widehat{x}_i$ and similarly $Y := \sum_i \beta_i x_i$, $\widehat{Y} := \sum_i \beta_i \widehat{x}_i$; then:*

$$\left|\langle X, Y\rangle - \langle \widehat{X}, \widehat{Y}\rangle\right| \leq \frac{\epsilon}{2}\left(\|X\|^2 + \|Y\|^2\right).$$

*Proof.* Since $\langle X, Y\rangle = \frac{1}{4}\left(\|X + Y\|^2 - \|X - Y\|^2\right)$, we can bound $A = \left|\langle X, Y\rangle - \langle \widehat{X}, \widehat{Y}\rangle\right|$ from above as:

$$\begin{aligned}
A &= \left|\frac{1}{4}\left(\|X + Y\|^2 - \|\widehat{X} + \widehat{Y}\|^2 - \|X - Y\|^2 + \|\widehat{X} - \widehat{Y}\|^2\right)\right| \\
&\leq \frac{1}{4}\left(\left|\|\widehat{X} + \widehat{Y}\|^2 - \|X + Y\|^2\right| + \left|\|\widehat{X} - \widehat{Y}\|^2 - \|X - Y\|^2\right|\right) \\
&\leq \frac{1}{4}\left(\epsilon \cdot \|X + Y\|^2 + \epsilon \cdot \|X - Y\|^2\right) \tag{8} \\
&= \frac{\epsilon}{4}\left(\|X + Y\|^2 + \|X - Y\|^2\right) \\
&= \frac{\epsilon}{2}\left(\|X\|^2 + \|Y\|^2\right),
\end{aligned}$$

where (8) follows from Lemma 4.1 with probability $1 - o(1)$, by our choice of $k$ (as Lemma 4.1 guarantees that each of $\|\widehat{X} + \widehat{Y}\|^2 = (1 \pm \epsilon)\|X + Y\|^2$ and $\|\widehat{X} - \widehat{Y}\|^2 = (1 \pm \epsilon)\|X - Y\|^2$ holds with probability $1 - o(1)$, and one can take a union bound over the two events). $\square$

**Lemma 4.3.** $H_{u,v} = 2M\langle \mathbf{r}_v - \mathbf{r}_u, \mathbf{r}_v - \mathbf{p}\rangle$ *where* $\mathbf{p} := \sum_u \pi_u \mathbf{r}_u$.

*Proof.* Consider the following expression of hitting times in terms of commute times by [43].

$$H_{u,v} = \frac{1}{2}\left[K_{u,v} + \sum_i \pi_i\left(K_{v,i} - K_{u,i}\right)\right]. \tag{9}$$

Dividing both sides of eq. (9) and using the relation $K_{u,v} = 2M\mathsf{Res}(u, v)$, we see that:

$$\begin{aligned}
\frac{H_{u,v}}{2M} &= \frac{1}{2}\left[\mathsf{Res}(u, v) + \sum_i \pi_i\left(\mathsf{Res}(v, i) - \mathsf{Res}(u, i)\right)\right] \\
&= \frac{1}{2}\left[\|\mathbf{r}_u - \mathbf{r}_v\|^2 + \sum_i \pi_i\left(\|\mathbf{r}_v - \mathbf{r}_i\|^2 - \|\mathbf{r}_u - \mathbf{r}_i\|^2\right)\right]. \tag{10}
\end{aligned}$$

Let's focus on the inner summation. After expanding out the squared norms, we see that:

$$\sum_i \pi_i \left( \|\mathbf{r}_v - \mathbf{r}_i\|^2 - \|\mathbf{r}_u - \mathbf{r}_i\|^2 \right) = \sum_i \pi_i \left( \|\mathbf{r}_v\|^2 - \|\mathbf{r}_u\|^2 \right)$$
$$- 2 \sum_i \pi_i \langle \mathbf{r}_v - \mathbf{r}_u, \mathbf{r}_i \rangle$$
$$= \left( \|\mathbf{r}_v\|^2 - \|\mathbf{r}_u\|^2 \right)$$
$$- 2 \langle \mathbf{r}_v - \mathbf{r}_u, \sum_i \pi_i \mathbf{r}_i \rangle$$
$$= \left( \|\mathbf{r}_v\|^2 - \|\mathbf{r}_u\|^2 \right) - 2 \langle \mathbf{r}_v - \mathbf{r}_u, \mathbf{p} \rangle.$$

Substituting this back into eq. (10), we can express $\frac{1}{2M} H_{u,v}$ as:

$$\frac{1}{2} \left( \|\mathbf{r}_v - \mathbf{r}_u\|^2 + \|\mathbf{r}_v\|^2 - \|\mathbf{r}_u\|^2 - 2 \langle \mathbf{r}_v - \mathbf{r}_u, \mathbf{p} \rangle \right)$$
$$= \|\mathbf{r}_v\|^2 - \langle \mathbf{r}_u, \mathbf{r}_v \rangle - \langle \mathbf{r}_v - \mathbf{r}_u, \mathbf{p} \rangle = \langle \mathbf{r}_v - \mathbf{r}_u, \mathbf{r}_v - \mathbf{p} \rangle. \qquad \square$$

**Lemma 4.4.** $|\widehat{H}_{u,v} - H_{u,v}| \leq 3\epsilon H_{\max}$.

*Proof.* Using Lemma 4.3, we see that

$$|\widehat{H}_{u,v} - H_{u,v}| = 2M \left| \langle \widehat{\mathbf{r}}_v - \widehat{\mathbf{r}}_u, \widehat{\mathbf{r}}_v - \widehat{\mathbf{p}} \rangle - \langle \mathbf{r}_v - \mathbf{r}_u, \mathbf{r}_v - \mathbf{p} \rangle \right|$$
$$\leq \epsilon M \left( \|\mathbf{r}_v - \mathbf{r}_u\|^2 + \|\mathbf{r}_v - \mathbf{p}\|^2 \right)$$
$$\leq 3\epsilon H_{\max},$$

where we used Corollary 4.2 in the first inequality and Definition 3.4 in the last inequality. $\qquad \square$

## D  Comparison: Effective Resistances vs. Shortest Path Distances

Given that effective resistance (ER) captures times associated with random walks in a graph, it is tempting to ask how effective resistances compare to shortest path distances (SPDs) between nodes in a graph. Indeed, for some simple graphs, e.g., trees, shortest path distances and effective resistances turn out to be identical. However, in general, effective resistances and shortest path distances behave quite differently.

Nevertheless, it is tempting to ask how effective resistance features compare to SPD features in GNNs, especially as there have been a number of recent model architectures that make use of SPD features (e.g., Graphormer [49], Position-Aware GNNs [52], DE-GNN [29]). We first note that the most natural direct comparison of our ER-based MPNNs with SPD-based networks does not quite make sense. The reason is that the analogous comparison would be to determine the effect of replace ERs with SPDs as features in our MPNNs. However, since our networks only use ER features along edges of the given graph, the corresponding SPD features would then be trivial (as the SPD between two nodes directly connected by an edge in the graph is 1, resulting in a constant feature on every edge)!

As a result, graph learning architectures that use SPDs typically either (a.) use a densely-connected network (e.g., Graphormer [49], which uses a densely-connected attention mechanism) that incurs $O(n^2)$ overhead, or (b.) pick a small set of *anchor nodes* or *landmark nodes* to which SPDs from all other nodes are computed and incorporated as node features (e.g., Position-Aware GNNs [52], DE-GNN [29]). We stress that the former approach generally modifies the graph (by connecting all pairs of nodes) and therefore does not fall within the standard MPNN approach, while the latter includes architectures that fall within the MPNN paradigm.

Furthermore, we note that DE-GNNs are arguably one of the closest proposals to ours, as they compute distance-encoded features. These features can be at least as powerful as our proposed affinity-based features *if* polynomially many powers of the adjacency matrix are used. However, for all but the smallest graphs, using this many powers will be impractical—in fact, [29] only use powers of $A$ up to 3, which would not be able to reliably approximate affinity-based features. We also

observe that the DE-GNN paper is concerned with learning representations of small sets of nodes (e.g., node-, link-, and triangle-prediction) and does not show how to handle graph prediction tasks, which the authors mention as possible future work. This makes a direct comparison of our methods with DE-GNNs difficult.

## D.1 Empirical Results

In an effort to empirically compare the expressivity of ER features with that of SPD features, we once again perform experiments on the PNA dataset, picking the following baselines that make use of SPD features:

- The first baseline is roughly an MPNN with *Graphormer-based features*. More precisely, it is a densely-connected MPNN with SPDs *from the original graph* as edge features. In order to retain the structure of the original graph, we also use additional edge features to indicate whether or not an edge in the dense (complete) graph is a true edge of the original graph. We also explore the use of the *centrality encoding* (in-degree and out-degree embeddings) from Graphormer as additional node features.
- The second baseline is the Position-Aware GNN (P-GNN), which makes use of "anchor sets" of nodes and encodes distances to these nodes.

The results of these baselines are shown in Table 7. In particular, we note that our ER-based MPNNs outperform all aforementioned baselines.

Table 7: Results on the PNA dataset for MPNNs with Graphormer-based features (yellow) as well as SPD-based P-GNNs (orange). Here, CE refers to the *centrality encoding*, which is incorporated in the relevant MPNNs as additional node features. Similarly, SPD refers to *shortest path distance* features — in the relevant MPNNs, shortest path distances between all pairs of nodes in the graph are incorporated as edge features, along with an additional edge feature indicating whether an edge exists in the input graph. Therefore, the MPNN baselines are all variants of the same model with additional node/edge features. Similarly, P-GNN [52] uses SPD features with respect to a set of chosen *anchor nodes*. The average score metric is, as before, the average of the $\log(MSE)$ metric over all six tasks, as in Table 1.

| Model | Average score |
|---|---|
| *MPNN + CE | -2.728 |
| *MPNN (dense) + SPD | -2.157 |
| *MPNN (dense) + CE + SPD | -2.107 |
| *P-GNN | -2.650 |
| **MPNN w/ resistive (edge) embeddings** | **-2.789** |
| **MPNN w/ all affinity measure features** | **-3.106** |

## D.2 Theory: ER vs. SPD

In addition to experimental results, we would like to provide some theory for why effective resistances can capture structure in GNNs that SPDs are unable to.

We will call an initialization function $u \mapsto \mathbf{h}_u^{(0)}$ on nodes of a graph *node-based* if it assigns values that are independent of the edges of the graph. Such an initialization is, however, allowed to depend on node identities (e.g., for the single-source shortest path problem from a source $s$, one might find it natural to define $\mathbf{h}_s^{(0)} = 0$ and $\mathbf{h}_u^{(0)} = +\infty$ for all $u \neq s$).

Consider the task of computing "single-source effective resistances," i.e., the effective resistance from a particular node to every other node. We show that a GNN with a limited number of message passing steps cannot possibly learn single-source effective resistances, even to nearby nodes.

**Theorem D.1.** *Suppose we fix $k > 0$. Then, given any node-based initialization function $\mathbf{h}_u^{(0)}$, it is impossible for a GNN to compute single-source effective resistances from a given node $w$ to any nodes within a $k$-hop neighborhood.*

*More specifically, for any update rule*

$$\mathbf{m}_{uv}^{(t+1)} = \psi_{t+1}\left(\mathbf{h}_u^{(t)}, \mathbf{h}_v^{(t)}, f_e(\mathbf{x}_{uv})\right)$$
$$\mathbf{h}_u^{(t+1)} = \phi_{t+1}\left(\mathbf{h}_u^{(t)}, f\left(\{\mathbf{m}_{uv} : v \in \mathcal{N}(u)\}\right)\right),$$

(11)

*there exists a graph $G = (V, E)$ and $u \in V$ such that after $k$ rounds of message passing, $h_v^{(k)} \neq$ $\mathsf{Res}(u, v)$ for some $v \neq u$ within a $k$-hop neighborhood of $u$.*

*On the other hand, there exists an initialization with respect to which $k$ rounds of message passing will compute the correct shortest path distances to all nodes within $k$-hop neighborhood.*

Note that the assumption on the initialization function in the above theorem is reasonable because enabling the use of arbitrary, unrestricted functions would allow for the possibility of precomputing effective resistances in the graph and trivially incorporating them as node features, which would defeat the purpose of computing them using message-passing.

*Proof.* Consider the following set of graphs, each on $4k + 1$ nodes:

Figure 2: Both of the above graphs are on $4k+1$ vertices, labeled $v_0, v_1, \ldots, v_{4k}$. The only difference is a single edge, i.e., the graph on the left has an edge between $v_{2k}$ and $v_{2k+1}$, while the one on the right does not have this edge.

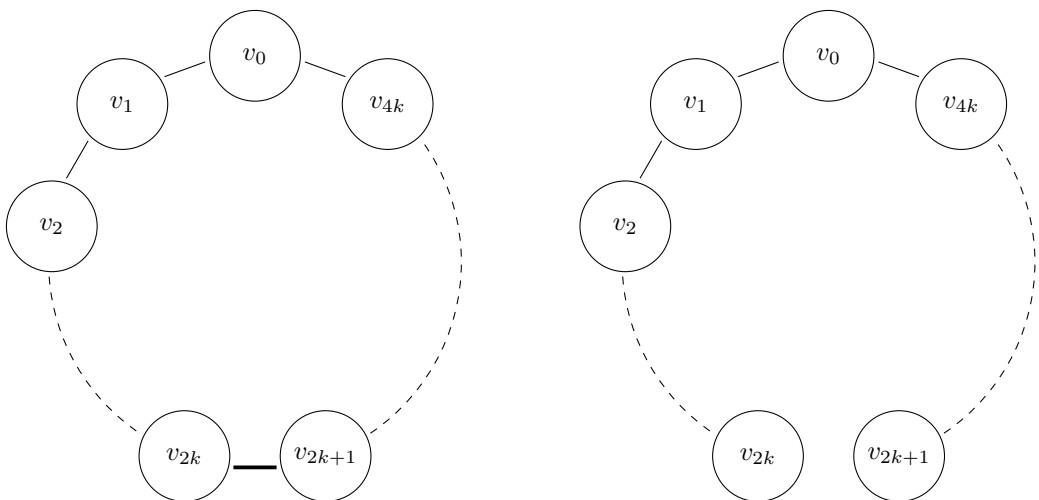

Let $V = \{v_0, v_1, \ldots, v_{4k}\}$. The first graph $G = (V, E)$ is a cycle, while the second graph $G' = (V, E')$ is a path, obtained by removing a single edge from the first graph (namely, the one between $v_k$ and $v_{k+1}$). Suppose the edge weights are all 1 in the above graphs.

Let $w = v_0$ be the source and let $\{\mathbf{h}_v^{(0)} : v \in V\}$ be a "local" node feature initialization. Note that for any GNN (i.e., update and aggregation rules in (11), add the formal update rule somewhere), the computation tree after $k$ rounds of message passing is identical for nodes $v_0, v_1, \ldots, v_k, v_{3k+1}, v_{3k+2}, \ldots, v_{4k}$ (i.e., the nodes within the $k$-hop neighborhood of $v_0$) in both $G$ and $G'$. This is because the only difference between $G$ and $G'$ is the existence of the edge between $v_{2k}$ and $v_{2k+1}$, and this edge is beyond a $k$-hop neighborhood centered at any one of the aforementioned nodes. Therefore, we will necessarily have that $\mathbf{h}_{v_i}^{(k)}$ is identical in both $G$ and $G'$ for $i = 1, \ldots, k, 3k + 1, 3k + 2, \ldots, 4k$.

However, it is easy to calculate the effective resistances in both graphs. In $G$, we have $\mathsf{Res}_G(v_0, v_i) = \frac{i(4k+1-i)}{4k+1}$, while in $G'$, we have $\mathsf{Res}_{G'}(v_0, v_i) = \min\{i, 4k + 1 - i\}$. Therefore, $\mathsf{Res}_G(v_0, v_i) \neq \mathsf{Res}_{G'}(v_0, v_i)$ for all $i = 1, 2, \ldots, k, 3k + 1, 3k + 2, \ldots, 4k$.

It follows that for any $i = 1, 2, \ldots, k, 3k + 1, 3k + 2, \ldots, 4k$, the execution of $k$ message passing steps of a GNN cannot result in $\mathbf{h}_{v_i}^{(k)} = \mathsf{Res}(v_0, v_i)$ for both $G$ and $G'$, which proves the first claim of the theorem.

For the second part (regarding single-source shortest paths), observe that single-source shortest path distances can, indeed, be realized via aggregation and update rules for a message passing network. In particular, for $k$ rounds of message passing, it is possible to learn shortest path distances of all nodes within a $k$-hop neighborhood. Specifically, for a source $w$, we can use the following setup: Take $\mathbf{h}_w = 0$ and $\mathbf{h}_u = \infty$ for all $u \neq w$. Moreover, for any edge $(u, v)$, let the edge feature $\mathbf{x}_{uv} \in \mathbb{R}$ simply be the weight of $(u, v)$ in the graph. Then, take the update rule (11) with $f_e, \psi_{t+1}$ as identity functions and

$$f_e(\mathbf{x}_{uv}) = \mathbf{x}_{uv}$$
$$\psi_{t+1}\left(\mathbf{h}_u^{(t)}, \mathbf{h}_v^{(t)}, f_e(\mathbf{x}_{uv})\right) = \mathbf{h}_u^{(t)} + \mathbf{x}_{uv}$$
$$f(S) = \min_S \{s \in S\}$$
$$\phi_{t+1}(a, b) = \min\{a, b\}.$$

It is clear that the above update rule simply simulates the execution of an iteration of the Bellman-Ford algorithm. Therefore, $k$ message passing steps will simulate $k$ iterations of Bellman-Ford, resulting in correct shortest path distances from the source $w$ for every node within a $k$-hop neighborhood. $\quad\square$

