# OpenReview forum: "Affinity-Aware Graph Networks"
_NeurIPS.cc/2023/Conference — NeurIPS 2023 poster_

### Official Review · Reviewer_2Foc · 2023-06-26

**Soundness:** 3 good
**Presentation:** 3 good
**Contribution:** 3 good
**Rating:** 7
**Confidence:** 3

**Summary:**

This work proposes to incorporate affinity measures as features into message-passing networks (MPNNs) in order to enhance the expressivity without enlarging the computational cost notably.
The authors introduce three examples of random-walk-based affinity measures, e.g., effective resistance, hitting, and commute times, and provide efficient computation and low-rank approximation for them.
On multiple graph benchmarks, ranging from graph-level tasks to transductive node-classification on large-scale graphs, the proposed affinity measures can remarkably improve the performance of MPNNs and reach state-of-the-art performance among models without privileged information.


**Strengths:**

1. According to my best knowledge, this work is the first to propose that incorporating affinity-based measures into MPNNs can improve the expressive power of MPNNs, going beyond the 1-WL algorithm.
2. This work also provides several techniques to improve the efficiency of the introduced affinity-based measures.
3. The authors show that MPNNs+affinity measures are strictly more powerful than the 1-WL algorithm via the example of regular graphs.
4. The empirical experiments demonstrate the effectiveness of the proposed methods on graph-level tasks and transductive node-level tasks. The experiment on large-scale graphs also shows the efficiency of the proposed technique.

**Weaknesses:**

1. The paper does not have more throughout comparisons with other positional encoding enhanced graph neural networks, theoretically and empirically. For details, please see the question section.
2. The baselines in empirical comparisons seem to have different numbers of parameters compared to the proposed method. It will be better to have a more detailed explanation on the choices of hyperparameters.











**Questions:**

(Loukas, 2020; Dwivedi et al., 2021) mention that encoding can enhance the ability of MPNNs to uniquely distinguish nodes and hence improve the expressive power of MPNNs.
I am curious whether the affinity measures enhance MPNNs in a similar way. Or they can provide extra valuable information that non-affinity-based encoding can not provide.
In other words, if there exists a non-affinity-based encoding able to distinguish isomorphic graphs as well as the proposed affinity measures, such as subgraph counting (Bouritsas et al., 2022), will incorporating the proposed affinity measures into MPNNs enjoy extra advantages over such an encoding? Whether graph-level tasks and node-level tasks have different conclusions regarding this?

If possible, it would be perfect if you could provide more comparisons theoretically/empirically against several PE-enhanced MPNNs (Dwivedi et al., 2021; Zhang et al., 2021; Wang et al., 2022; Lim et al., 2022; Li et al., 2020; Bouritsas et al., 2022) and potentially the PE for Graph Transformers (Zhang et al., 2023; Ma et al., 2023).




- Loukas, A. (2020). How hard is to distinguish graphs with graph neural networks? Adv. Neural Inf. Process. Syst.
- Dwivedi, V. P., Luu, A. T., Laurent, T., Bengio, Y., & Bresson, X. (2021). Graph Neural Networks with Learnable Structural and Positional Representations. Proc. Int. Conf. Learn. Representations.
- Bouritsas, G., Frasca, F., Zafeiriou, S. P., & Bronstein, M. (2022). Improving Graph Neural Network Expressivity via Subgraph Isomorphism Counting. IEEE Transactions on Pattern Analysis and Machine Intelligence, 1–1.
- Zhang, Z., Cui, P., Pei, J., Wang, X., & Zhu, W. (2021). Eigen-GNN: A Graph Structure Preserving Plug-in for GNNs. IEEE Transactions on Knowledge and Data Engineering.
- Wang, H., Yin, H., Zhang, M., & Li, P. (2022). Equivariant and Stable Positional Encoding for More Powerful Graph Neural Networks. Proc. Int. Conf. Learn. Representations.
- Lim, D., Robinson, J. D., Zhao, L., Smidt, T., Sra, S., Maron, H., & Jegelka, S. (2022). Sign and Basis Invariant Networks for Spectral Graph Representation Learning. ICLR 2022 Workshop on Geometrical and Topological Representation Learning.
- Li, P., Wang, Y., Wang, H., & Leskovec, J. (2020). Distance Encoding: Design Provably More Powerful Neural Networks for Graph Representation Learning. Adv. Neural Inf. Process. Syst.
- Zhang, B., Luo, S., Wang, L., & He, D. (2023). Rethinking the Expressive Power of GNNs via Graph Biconnectivity. Proc. Int. Conf. Learn. Representations.
- Ma, L., Lin, C., Lim, D., Romero-Soriano, A., K. Dokania, Coates, M., H.S. Torr, P., & Lim, S.-N. (2023). Graph Inductive Biases in Transformers without Message Passing. Proc. Int. Conf. Mach. Learn.








**Limitations:**

No limitations are provided.

---

> ### Author Rebuttal · Authors · 2023-08-10
>
> > 1. The paper does not have more throughout comparisons with other positional encoding enhanced graph neural networks...
> >  If possible, it would be perfect if you could provide more comparisons...
>
> We note that we have already provided comparisons to the GSN model of Bouritsas et. al. for ogbg-molhiv.
>
> Regarding the DE-GNN model of Li et al., we note that it is difficult to compare our method with theirs, as their paper does not handle graph prediction tasks. In fact, the DE-GNN paper is concerned with learning structural representations of small sets of nodes and notes: To quote the paper, _“Our approaches may also help other tasks based on structural representation learning, such as graph-level classification/regression [14, 16, 23, 25, 30] and subgraph counting [59], which we leave for future study.”_ The fact that most experiments in our paper concern graph prediction makes a proper comparison with DE-GNN elusive. However, we have nevertheless attempted to compare our work to the key ideas of Li et al. and have included comparisons to distance-based features in Appendix D.
>
> Below, we provide some additional comparisons to PE-enhanced MPNNs:
>
> Dwivedi et al., 2021: On ogbg-molpcba, we note that the authors’ PNA-LSPE model (on 4 layers) obtains a test AP of 28.4 ± 0.2, while their GatedGCN-LSPE model obtains a test AP of 26.7 ± 0.2. We note that our best 4-layer, 8-layer, 16-layer models are competitive with the former (within the margin of error) while surpassing the latter.
>
> > 2. The baselines in empirical comparisons seem to have different numbers of parameters compared to the proposed method. It will be better to have a more detailed explanation on the choices of hyperparameters.
>
> When we add scalar features (e.g., ER or HT), there are only 2-3 extra features added, so this does not appreciably increase the number of hyperparameters.
>
> Even when we add the vector resistive embeddings in addition to the scalar features, we find that the number of hyperparameters increases by less than 10% (e.g., for the PNA dataset, the comparable MPNN models with and without affinity measures have around 9.7M and 8.9M hyperparameters, respectively, representing a 9% increase).
>
> Regarding the choice of hyperparameters, we find that using all affinity measures together (both ER and HT scalar features as well as vector resistive embeddings) generally works the best; however, on larger datasets for which GPU memory is a bottleneck, we stick to using just the scalar features.
>
> > Q. (Loukas, 2020; Dwivedi et al., 2021) mention that encoding can enhance the ability of MPNNs to uniquely distinguish nodes...
>
> Appendix D shows some experiments comparing affinity measures to other positional/structural encodings such as shortest path distance encodings and centrality encoding (which has been used in graph transformer architectures such as Graphormer (Ying et al. 2021)), providing evidence that affinity-based encodings can provide extra information that other encodings cannot. We further note that the experiments in Appendix D are on a collection of graph-level and node-level tasks, providing indication that affinity-based encodings can be beneficial in both settings.
>
> We nevertheless note that the question of whether affinity-based encodings can provide advantages via approaches other than feature augmentation is a scintillating one. While we have primarily incorporated affinity measures as features to an MPNN, one need not be limited by this route. Positional encodings (that allow nodes to be located/distinguished within a graph) have proved to be particularly important for non-message passing architectures such as Graph Transformers, which crucially rely on positional and structural encodings to make up for the loss of inductive bias arising from the separation of the computation graph from the input graph. A fascinating direction for future work would be to explore the use of affinity-based positional encodings in such architectures.

---

> > ### Comment · Reviewer_2Foc · 2023-08-16
> >
> > Thank you for your responses.
> > I will raise my score to 7.

---

> > > ### Author Response · Authors · 2023-08-17
> > > **Thank you!**
> > >
> > > Thank you! We appreciate your taking the time to read through our rebuttal and adjust your score upwards.

---

### Official Review · Reviewer_vQdB · 2023-07-03

**Soundness:** 3 good
**Presentation:** 3 good
**Contribution:** 3 good
**Rating:** 7
**Confidence:** 4

**Summary:**

The paper proposes a strategy to strengthen the node and edge features in order to enhance the expressiveness of a message passing neural network. In particular, the paper introduces a set of effective resistance (ER) features, including node-level resistive embedding, which further derives two edge-level affinity measures: hitting time and commuting time/effective resistance. Additionally, the paper employs Constructive Johnson-Lindenstrauss to randomly project resistive embeddings to a lower-dimensional space, which is utilized to expedite the computation of hitting time and commuting time. Finally, the paper present the empirical evaluations on the proposed ER features.

**Strengths:**

1. The paper is well-written: the motivation is clearly presented, and most sections are straightforward to follow.
2. The fast approximation of computing hitting time and commuting time is novel. The dimensionality of resistive embedding is reduced without the requirement of specifying a set of anchor nodes.
3. The experiment on large graphs demonstrates that the computation cost of adding these features to MPNN is manageable.

**Weaknesses:**

1. The paper would benefit from providing stronger evidence to support certain claims made throughout the text. Please refer to the "Questions" section for details.

**Questions:**

1. I would like to seek clarification from the authors regarding the connection between "1-WL" and "standard GNN" in the context of node classification, as demonstrated in the paper to support the claim made by Theorem 3.6. The provided example gives me the impression that a "standard GNN" represents each node using a spanning tree rooted at the node of interest (referred to as a computation tree) and distinguishes nodes based on the isomorphism of these spanning trees.

   To facilitate readers' understanding of this concept, I kindly request the authors to provide a specific example of the computation tree that a standard GNN employs to represent a node. It would also be helpful if the authors could provide some justification or reference regarding the connection between the node representation learned by standard GNNs and the utilization of spanning trees.

2. By utilizing the initial node feature as the one-hot encoding of the node's index, denoted as $\textbf{h}_v^{(0)} = \textbf{e}_v$, and employing the forward function $\mathbf{H}^{(t+1)} = \mathbf{A}\mathbf{H}^{(t)}$, we can observe that after 4 layers, the node embeddings generated by the standard GNN successfully differentiate the 3 color categories. Specifically, if we define a score $s_v = <\textbf{h}_v^{(4)}, \textbf{e}_v>$, the value for nodes {1,2} is 15, for nodes {3,4,7,8} it is 17, and for nodes {5,6} it is 19. This observation contradicts the claim that "a standard GNN that is limited by the 1-WL test cannot distinguish any pair of nodes in a regular graph." Consequently, I have some follow-up questions based on this discrepancy.

   (1) What is the specific constraint on the input features to a standard GNN?

   (2)  Will the benefit from resistive embedding become less evident as we increase the depth of an MPNN?

3. In Appendix D.2, a comparison is conducted between resistive embedding and shortest-path-distance embedding in terms of their ability to differentiate nodes within a cycle and the corresponding nodes in the path obtained by removing one edge from the cycle.
The example effectively illustrates that node embeddings based on shortest-path-distance are insufficient in capturing the divergence that can be successfully captured by resistive embeddings. However, this raises a valid concern regarding real-life networks where edges may be lost due to noise. In such cases, will the strong ability of resistive embeddings to distinguish between graphs that are close in terms of edit distance potentially make them susceptible to overfitting the noise?

---

> ### Author Rebuttal · Authors · 2023-08-10
>
> > 1. I would like to seek clarification from the authors regarding the connection between...
>
> Indeed, the message passing mechanism of a GNN gives rise to a computation tree with depth given by the number of message passing steps. As an example, for the graph in Figure 1, if we use two rounds of message passing (i.e., 2 MPNN layers), the resulting node representation for node 1 would arise from the following depth-2 computation tree rooted at 1:
>
> ```
> 1
> ├── 2
> │   ├── 1
> │   ├── 3
> │   └── 4
> ├── 7
> │   ├── 1
> │   ├── 6
> │   └── 8
> └── 8
>     ├── 1
>     ├── 5
>     └── 7
> ```
>
> We would be happy to provide any further examples or clarification if needed during the discussion phase and can incorporate them in the camera-ready version.
>
>
> > 2. By utilizing the initial node feature as the one-hot encoding of the node's index...
>
> The standard setting for comparing GNNs to the 1-WL test is the one in which all nodes start with the same representation/color before the series of updates. Therefore, in the case of a regular graph, if one starts with the same representation for every node, updates (whether aggregation in a GNN or color/hashing updates in 1-WL) will leave every node with the same representation after every step.
>
> You note correctly that starting with one-hot encodings of the node index as initial node features will allow nodes to be distinguished by a GNN; however, in this case the appropriate comparison is the 1-WL procedure which _also starts with one-hot encodings as initial ‘colors’_, which allows all nodes to be trivially distinguished from each other. The fact that node indices can improve GNN expressivity has been well documented: even random node features (essentially a more scalable way of providing node indices, as one typically does not want the node feature dimensionality to scale with the number of nodes n, as one-hot encodings of nodes would do) improve GNN expressivity (see [1], [2]).
>
> Specifically, we answer your questions as follows:
>
> 1. In the “standard” setting, one starts with the same representation for every node (i.e., no use of distinguishing node features). However, one can also start with more general node representations/colorings, but in this case the 1-WL test being compared to must also use the same initial node colorings.
> 2. Increasing the depth of an MPNN too much can pose various general issues, such as vanishing gradient or oversquashing (see [3], [4], [5]). We expect these to still be the case when using resistive embeddings, so the benefit for large depths could be muted, as is the case more broadly for MPNNs.
>
> [1] Ryoma Sato. “A Survey on The Expressive Power of Graph Neural Networks.”
>
> [2] Ryoma Sato, Makoto Yamada, Hisashi Kashima. “Random Features Strengthen Graph Neural Networks.”
>
> [3] Uri Alon, Eran Yahav. “On the Bottleneck of Graph Neural Networks and its Practical Implications.”
>
> [4] Jake Topping, Francesco Di Giovanni, Benjamin Paul Chamberlain, Xiaowen Dong, Michael M. Bronstein. “Understanding over-squashing and bottlenecks on graphs via curvature.”
>
> [5] Francesco Di Giovanni, Lorenzo Giusti, Federico Barbero, Giulia Luise, Pietro Lio', Michael Bronstein. “On Over-Squashing in Message Passing Neural Networks: The Impact of Width, Depth, and Topology.”
>
>
> > 3. In Appendix D.2, a comparison is conducted between resistive embedding and shortest-path-distance embedding...
>
> In a worst-case scenario, removing an edge can change the effective resistance between a given pair of nodes appreciably. However, this is not a phenomenon limited to effective resistance—even in the case of shortest path distance, adding/removing a single edge can change the distance between the endpoints by an arbitrarily large factor.
>
> Nevertheless, we observe that in **real-world networks**, effective resistance tends to be a fairly _robust_ measure. Note that removing an existing edge $e$ can never decrease any effective resistances, and for a connected graph, one can show that it can only increase the effective resistance between any pair of nodes by at most a factor of $1/(1-R_e)$. (Note that we necessarily have $R_e \leq 1$ for any existing edge of the graph, and as long as e is not a bridge edge whose removal disconnects the graph, the inequality is strict.) Applying this fact to the underlying graph of the ogbn-arxiv dataset, we observe that 95% of the edges in the graph have $R_e < 0.5$, which implies that the removal of any such edge will increase the effective resistance between any pair of nodes by a factor of $< 2$ (this property does not, however, hold for shortest path distance). Similarly, it can be shown that adding an edge $(u,v)$ can only decrease the effective resistances by a factor of $1/(1+R_{uv})$. Thus, adding an edge between nearby nodes will also not affect the effective resistances much.

---

> > ### Comment · Reviewer_vQdB · 2023-08-20
> > **Thank you for rebuttal**
> >
> > hank you for your feedback. I appreciate your consideration in including the discussions we've had in the final revisions of the paper. The paper presents an intriguing and engaging topic, and based on its quality, I highly recommend its acceptance. I also want to acknowledge that I have upgraded my score to 7. Best of luck!

---

### Official Review · Reviewer_YKRe · 2023-07-06

**Soundness:** 3 good
**Presentation:** 2 fair
**Contribution:** 3 good
**Rating:** 4
**Confidence:** 3

**Summary:**

This paper proposes to use affinity measures as additional features that can incorporate in common standard MPNNs and theoretically and empirically shows the improvement in expressiveness and performance in some datasets without loss of much scalability. The main contributions are listed below.
1. Generality. The proposed approach can be adopted in standard MPNNs.

2. Expressiveness. Incorporating the proposed features can make standard MPNNs more powerful than the 1-WL test.

3. Scalability. The affinity measures used in this paper are based on the random walk. By approximation, the features derived from the proposed measures can be incorporated with high scalability.

4. Performance. MPNNs equipped with the proposed additional features can make progress or even rank 1 in some datasets.

**Strengths:**

1. This paper was inspired by effective resistances and proposed HT, ER and resistive embeddings all help common MPNNs in both performance and expressiveness. As a feature-based approach, it can adapt to many standard MPNNs and does not need to specify any anchor or target nodes to capture both distance and connectivity information using general-purpose node and edge features.

2. By Approximation and fast computation of Resistive Embeddings and HT, the proposed features can scale up to large graphs in time and space complexity.

3. The experimental results are impressive. With relatively low computation cost, the proposed method can have a competitive performance on small datasets and higher performance on homogeneous/heterogeneous citation networks. Further, the best performance achieved on PCQM4M-v1 is impressive.

4. Both theoretical proof and empirical evaluations are provided.

**Weaknesses:**

1. The motivation needs to be clarified more clearly. What the main drawbacks of the existing approaches are and which the proposed one surmounts? It’s said that the proposed approach “without the need for specifying any anchor or target nodes” and the scalability problem is mentioned in the experiments section. However, the advantages should be clarified further and organized better since there are other feature-based works and other categories of works aiming at a similar goal. Some practical evaluation or comparison may help if possible.

2. To demonstrate the superiority and necessity of the proposed affinity feature-based approach, other similar approaches need to be compared. Similar to the proposed affinity-based features, approaches incorporating other additional features (e.g. Random features or other dedicated designed features), and other methods that can improve GNNs’ expressiveness (e.g. other 2 directions mentioned in Expressivity para in Sec. 2) should also be compared in scalability and performance.

3. Although by approximation, the computation of random embeddings can be done in near-linear time and can be scaled to a large dataset, it’s better to provide the actual runtime of the proposed and baseline methods.

4. Theorem 3.6 claims that MPNNs that make use of any one of the proposed features are strictly more powerful than the WL-1 test. However, the experiments for it are not enough. Experiments on ogbg-molhiv show that it outperforms substructure counts, but the reasons are not clear. Experiments on some (artificial) datasets (e.g. [1][2]) which are designed for testing expressiveness would be better.

5. More well-organized writing should be considered. The notations scattering in Sec. 3.1-3.3 bring some difficulty in reading. The different baselines, settings, and experimental results in each subsection in Sec. 5 are scattered, either in the tables or in the text, which may weaken the empirical demonstration effects of experiments and cause some confusion. Some nouns or notations are introduced but not explained clearly, such as ER scalar features, node or edge features used in experiments, e_v in Eq. (1), and random rotations. Introducing them in a centralized and unified manner may help.

**Questions:**

1. It’s said in Sec. 5.4 that ER embedding dimension become impractically high and authors alter to use ER scalar features. What are the theoretical and empirical (experimental) differences between the two kinds of features? If ER scalar features in small graphs can make the same progress with ER embedding? If not, why?

2. Incorporating hitting time features, resistive embeddings, and effective resistance are all proven to be more powerful than 1-WL in Sec. 3.4. Why does using ER/HT in experiments have different results? Is there any suggestion about selecting which feature?

3. The performance of the proposed method does not surpass the DGN in Table 2 and does not show advantages when more layers are in Table 3. More explanations are expected although the proposed method shows advantages in other aspects.

4. How the proposed features help in heterogeneous networks may be interesting. Possible explanations are expected.

**Limitations:**

1. As said in W3, the actual runtime in large graphs may be much higher than (scalable) baseline methods.

2. As said in Q2, experiments use different features (ER embedding/HT …), thus suggestion about selecting feature is needed.

3. When demonstrating the expressiveness of the proposed approach, experiments on some datasets tailored for evaluating expressive power are expected. For instance, one can synthesize a dataset containing instances indistinguishable by 1-WL(e.g. [1][2]) to see if the proposed approach can distinguish.

[1] Sato R, Yamada M, Kashima H. Random features strengthen graph neural networks[C]//Proceedings of the 2021 SIAM international conference on data mining (SDM). Society for Industrial and Applied Mathematics, 2021: 333-341.
[2] Abboud R, Ceylan I I, Grohe M, et al. The surprising power of graph neural networks with random node initialization[J]. arXiv preprint arXiv:2010.01179, 2020.

---

> ### Author Rebuttal · Authors · 2023-08-10
>
> > 1. The motivation needs...
>
> Features in GNNs are typically subject to a tradeoff between (a.) efficiency and (b.) high expressivity. For instance, random features or node index features address (a.) at the cost of (b.), while other features such as substructure counts help with (b.) but fall short in (a.), e.g., computation can be superquadratic in the number of nodes. Our work is _motivated by this tradeoff and addresses the challenge of bridging (a) and (b)_. We highlight that our approach is also general in nature, something often not true for structural features where one needs to identify the (domain-specific) structures of interest.
>
> > 2. To demonstrate...
>
> We note that the submission includes some comparisons to approaches you mention. For the PNA dataset, the results in Table 1 include MPNN baselines with (a) random features and (b) DGN features. Furthermore, for the molecular datasets (ogbg-molhiv, ogbg-molpcba), we have included comparisons to both the junction tree-based HIMP baseline (mentioned as [13] in Section 2) as well as the substructure-based GSN baseline ([6] in Section 2). We have additionally included random walk-based baselines such as GRWNN (see Table 2).
>
> > 3. Although by...
>
> Computation times for the ogbn-arxiv dataset are provided on page 8. We further highlight that the computation of effective resistances proceeds via computation of the embeddings, followed by a quick computation of L2-squared distances between pairs of embeddings (see Lemma 3.2). Without the use of approximate random embeddings, the computation would require an expensive matrix inversion that would be infeasible for the size of the dataset (169,343 nodes).
>
> > 4. Theorem 3.6 claims...
>
> Thank you for the suggestion. As mentioned in the global response to all reviewers, we have now included results on the synthetic datasets, TRIANGLE(N) and LSC(N) in [1]. Once again, the results (test ROC-AUC score, higher is better) are as follows (see global response for experimental details):
>
> _TRIANGLE (N)_:
>
> GIN:
> - GIN: 0.5
> - GIN + rand feats: 0.924
> - **GIN + res embeddings: 0.911**
>
> GCN:
> - GCN: 0.5
> - GCN + rand feats: 0.857
> - **GCN + res embeddings: 0.879**
>
> _LSC (N)_:
>
> GIN:
> - GIN: 0.5
> - GIN + rand feats: 0.847
> - **GIN + res embeddings: 0.935**
>
> GCN:
> - GCN: 0.5
> - GCN + random features: 0.794
> - **GCN + res embeddings: 0.873**
>
> > 5. More well-organized writing...
>
> We will work on improving the organization and writing in the final version.
>
> > Q1.
>
> There are two potential sources of scalability bottlenecks: **computational complexity** and **GPU memory**. The impracticality referred to in Sec. 5.4 concerns the latter, not the former. Indeed, even on ogbn-arxiv, we are able to compute resistive embeddings without any problem and, in fact, use them to obtain ER scalar values (via Lemma 3.2) to **avoid computationally infeasible matrix inversions**. However,  the high dimensionality of the embeddings makes them infeasible to use in GPU memory.
>
> We note that resistive embeddings are still richer than ER/HT scalar values (the latter can be computed from the former) and have observed empirical improvements on the datasets where we try them. In addition to Table 1 for the PNA dataset, which shows lift from using embeddings along with scalar features, we note that for ogbg-molhiv, using ER scalar features yields a Test % ROC-AUC of 77.75 ± 0.426, while using hitting time scalar features yields 76.56 ± 0.915. However, as reported in Table 2, additionally _using embeddings produces a much higher Test % ROC-AUC of 79.13 ± 0.358_.
>
> > Q2.
>
> While it is indeed true that any of the features (ER, HT, resistive embeddings) results in more power than 1-WL, it is not _a priori_ clear that including these features would necessarily give better empirical test accuracy. We nevertheless show that affinity measures do improve empirical performance to varying degrees. It should also be noted that we generally use the scalar features as _edge features_, while embeddings are incorporated as both _node features_ and _edge features_. This can result in differing test performance, as MPNNs use node and edge features differently.
>
> Regarding selection of features, we have generally found that _using both embeddings and scalar features (ER and HT) results in the best performance_. However, for very large graphs (thousands of nodes), we stick to using just scalar features due to GPU memory bottlenecks.
>
> > Q3.
>
> Approaches to improve expressivity and performance of GNNs fall into three categories, viz., feature augmentation, message passing modulation, and graph modification — our technique falls under the first approach, while DGN falls under the second. While our approach has been to augment affinity measures as features, one is not by any means limited to this approach — an exciting future direction would be to use affinity measures to modulate the message passing, which would provide a fairer comparison to DGN. In order to give some indication of an apples-to-apples comparison to DGN, in Table 1 we have provided a “DGN (features)” baseline, which (instead of modulating the message passing mechanism of the MPNN) incorporates the relevant features via feature augmentation.
>
> > Q4.
>
> Heterogeneous networks may be viewed as graphs with multiple node and/or edge types. One can apply our techniques based on affinity measures in a few different ways. One way is to consider all types of nodes/edges together and compute affinity measures on the combined graph in the usual way, then incorporating them in the desired message-passing architecture (this is, for instance, our approach for experimentation on ogbg-molhiv, which has multiple edge types arising from different bond types). Another way is to consider the various node or edge types separately and compute affinity measures for separate graphs in isolation. Furthermore, with either approach, one can optionally aggregate different edge types separately, e.g., using an architecture like R-GCN).

---

> > ### Comment · Reviewer_YKRe · 2023-08-21
> >
> > Thanks for the author's response, and it partially addressed my concerns.

---

### Official Review · Reviewer_8Ddc · 2023-07-07

**Soundness:** 3 good
**Presentation:** 3 good
**Contribution:** 2 fair
**Rating:** 5
**Confidence:** 3

**Summary:**

In this paper, the authors present MPNN using affinity measures as node and edge features. As the affinity measures, the authors propose effective resistance, hitting times, and resistive embeddings. The authors demonstrate the effectiveness of using the affinity measures through experiments on various datasets. Experimental results on large graph datasets prove the good scalability of the scheme.

**Strengths:**

1. I considered the need for a comparison with shortest path distances (SPDs), however, the related description in Section D (in the supplementary material) is convincing.

2. The proposed scheme has low computational complexity and good scalability.

3. The paper is well-written and easy to understand.

**Weaknesses:**

It appears that the setting for MPNN models used in the experiments is not fair. According to Table 6 in the supplementary material, the number of layers is set to 3 for all models. Do the authors consider this setting to be fair? Furthermore, the hidden size of each layer is set differently for each method. What is the basis for setting the hidden size for each method? For example, the hidden size for GAT is set to 64 while that for HT + ER (rand rot) is set to 512.

minor comment
- fom -> from (Line 112)

**Questions:**

Q1. Do the authors think that the experiments with the current hyper-parameter setting are fair?

Q2. Could the authors explain on what basis they set the hyper-parameters for each method?

Q3. Is it possible to improve performance by adding the proposed features to DGN?

**Limitations:**

The authors do not describe the potential negative social impact of their work while present future work to develop the work.

---

> ### Author Rebuttal · Authors · 2023-08-10
>
> > Q1. Do the authors think that the experiments with the current hyper-parameter setting are fair?
>
> We note that the hyperparameters for each model/baseline were chosen via a hyperparameter sweep. This includes, for instance, the hidden size. For each model in the table, we provide the best setting of hyperparameters, i.e., the setting that results in the best score. Therefore, we believe that the experiments do indeed provide a fair comparison. We thank the reviewer for raising this important question, and we will make sure to clarify these experimental details regarding the hyperparameter sweep.
>
> > Q2. Could the authors explain on what basis they set the hyper-parameters for each method?
>
> We swept over hyperparameters as follows:
>
> - Hidden size: {64, 128, 256, 512}
> - Learning rate: {1e-4, 1e-3, 5e-3}
> - Number of layers (for MLP): {2, 3}
> - Message passing steps: {1, 2, 3}
>
> All combinations of the above hyperparameters were tried out (with repetition over multiple seeds) for every model/baseline, and the best setting of hyperparameters was reported for each model.
>
> We will, once again, add further clarification to the paper on the details of the hyperparameter sweep.
>
> > Q3. Is it possible to improve performance by adding the proposed features to DGN?
>
> Indeed, it is possible to add affinity features to DGN. We have not tested this out here, as DGN requires significantly altering the model architecture by modulating the message passing mechanism. However, in order to give some indication of how affinity measures can work in tandem with the features considered in DGN, below we report a new set of experimental results on the PNA dataset for an MPNN model that uses affinity features along with  _DGN features_ (note that use of _DGN features_ without affinity features was already reported in Table 1 of the submission):
>
> - **DGN features + affinity measures: -2.938**
> - DGN features only (already reported in Table 1): -2.743
>
> As in the paper, the above results on the PNA dataset denote the average score (log(MSE)) over the relevant set of six tasks, and the results are averaged over 10 random seeds. The above numbers show that adding affinity measures results in an improvement over _DGN features_.

---

> > ### Comment · Reviewer_8Ddc · 2023-08-20
> > **Rebuttal Acknowledgement**
> >
> > Thank you for the response. These comments substantially address my concerns.

---

### Author Rebuttal · Authors · 2023-08-10

We thank the reviewers for taking the time to look through our submission and provide insightful comments and questions. In light of the comments raised by reviewers, we have provided further results and addressed several concerns. We respond to each reviewer individually.

There is one **experimental update** that we would like to highlight to all the reviewers:

In response to questions about expressivity, we are now providing new experimental results with comparisons to the synthetic datasets, _TRIANGLE(N)_ and _LSC(N)_ (from [1]) to show stronger evidence that affinity measures exhibit improved expressivity. For both datasets, we follow the setup in [1] and use GIN and GCN models, which we augment with resistive embedding node features and compare to both (a.) vanilla models and (b.) models augmented with random features (the main contribution of [1]). The results (test ROC-AUC score, higher is better) are as follows:


_TRIANGLE (N)_:

GIN:
- GIN: 0.5
- GIN + random features: 0.924
- **GIN + resistive embeddings: 0.911**

GCN:
- GCN baseline: 0.5
- GCN + random features: 0.857
- **GCN + resistive embeddings: 0.879**


_LSC (N)_:

GIN:
- GIN baseline: 0.5
- GIN + random features: 0.847
- **GIN + resistive embeddings: 0.935**

GCN:
- GCN baseline: 0.5
- GCN + random features: 0.794
- **GCN + resistive embeddings: 0.873**


We find that using resistive embeddings results in the highest ROC-AUC scores in all cases except for GIN on TRIANGLE, in which case random features and resistive embeddings are comparable. This provides strong evidence that the given affinity measures indeed provide enhanced expressivity.

[1] Sato R, Yamada M, Kashima H. Random features strengthen graph neural networks[C]//Proceedings of the 2021 SIAM international conference on data mining (SDM). Society for Industrial and Applied Mathematics, 2021: 333-341.

---

### Decision · Program_Chairs · 2023-09-21

**Decision:**

Accept (poster)

**Comment:**

This paper provides a steady technical contribution to graph neural networks. Many of the concerns raised by the reviewers were addressed through discussions with the authors.